# First Arrival Picking of Zero-Phase Seismic Data by Hilbert Envelope Empirical Half Window (HEEH) Method

**DOI:** 10.3390/s22197580

**Published:** 2022-10-06

**Authors:** Amen Bargees, Abdullatif A. Al-Shuhail

**Affiliations:** Geosciences Department, King Fahd University of Petroleum and Minerals, Dhahran 31261, Saudi Arabia

**Keywords:** empirical rule, first arrival travel time picking, Hilbert transform, zero-phase wavelet

## Abstract

First arrival travel time picking is an important step in many seismic data-processing applications. Most first arrival picking methods search for a sudden jump in seismic energy at trace onsets, which is clearly appropriate for minimum-phase data. This paper proposes a method for the first arrival picking of non-minimum phase data based on complex trace analysis. The Hilbert integral transform generates a complex seismic trace, followed by extraction of the envelope. The first arrival identification introduces an outlier detection method that uses the widely used three-sigma rule of thumb, which is commonly used in most software algorithms to identify outliers. The proposed method ultimately generates logical windows of ones (at the locations of outliers) and zeros (elsewhere). The first arrival is selected in the middle of the first outlier window. Testing the proposed method on zero-phase synthetic data with added 10% and 20% random noise, the method detected the true first arrivals accurately. Furthermore, tests on real Vibroseis data showed that the method recognizes the first arrivals with 67% accuracy within 20 milliseconds of their corresponding arrival times manually picked by an experienced geophysicist.

## 1. Introduction

First arrival travel time picking is essential for near-surface model building. The general methods implemented typically fall into two categories, namely manual and automatic. However, both methods encounter problems in cases of complex near-surface geology, some types of seismic sources, and poor signal-to-noise ratio (SNR) [1]. A manual approach requires visual inspection of each seismic trace and the selection of the appropriate location for the first arrival. This process is feasible with a small dataset or synthetic data, but when it comes to real-life situations, it has its limitations because of the large volumes of seismic data involved, noisy seismic traces, the need for experienced personnel, and personal bias. The automatic first arrival picking approach works better under distinct circumstances where no method is valid for all datasets and, thus, needs careful consideration. Peraldi and Clement (1972) addressed the first arrival travel time using cross-correlation of adjacent traces [2]. Numerous methods, such as short-term average/long-term average (STA/LTA) [3], Akaike information criterion ([4,5]), instantaneous travel time [6], edge detection [7], polarization analysis [8], neural networks ([9,10,11]), and deep learning algorithms [12], have been developed and improved to pick the first arrivals of seismic traces.

We propose a new technique for selecting the first arrival travel time of a zero-phase shot record starting from the Hilbert transform, followed by the envelope, which produces a noticeable jump at the first arrival travel time. Although the jump is a distinct criterion for first arrival travel time picking, the user has yet to pinpoint the exact first arrival pick location, either at the beginning, center, or even the end of the jump, depending on the wavelet phase. This is followed by an outlier identification step using the 68–95–99.7 criterion, also known as the empirical rule or the three-sigma rule of thumb [13]. The empirical rule generates a logical window at the first arrival travel time. For zero-phase wavelets, the midpoint (half) of the first outlier window indicates the first arrival time. This sequence of steps, including Hilbert, envelope, empirical, and half-window steps, defines our proposed HEEH method as the first arrival travel time without human bias. The code used can be checked in the Appendix A. Using the proposed method, tests were conducted on synthetic and real-shot records.

## 2. Materials and Methods

We illustrate the steps of the proposed method on a real Vibroseis trace (Figure 1). Figure 2 shows the results of applying the four main steps of the HEEH workflow to a Vibroseis trace.

### 2.1. Hilbert Transform

The Hilbert transform is a linear operator calculated by convolving a function with the operator 1/πt. In the frequency domain, the Hilbert transform simply adds a phase shift of 90° to the phase spectrum of the Fourier transform. The application of the Hilbert transform to a seismic trace, *x*(*t*), produces a complex seismic trace *z*(*t*), as follows:
*z*(*t*) = *x*(*t*)+ *i**y*(*t*) (1)
where *y*(*t*) is the seismic trace rotated by 90° produced by the Hilbert transform [14]. The envelope is calculated as follows:(2)$$a\left(t\right)=\sqrt{x^{2}\left(t\right)+y^{2}\left(t\right)}$$

The HEEH method requires further adjustments to automate and avoid human biases. Therefore, the jump needs to be further defined, as the change is rarely abrupt. The first arrival travel time can be at the beginning, center, or end of the break, depending on the type of source wavelet.

### 2.2. Empirical Rule

The empirical rule is a statistical technique that provides information on the magnitude of the deviation between the values of observations in a dataset. The empirical rule states that “for a normal distribution, 68% of data will fall within the first standard deviation, 95% within the first two standard deviations, and 99.7% within the first three standard deviations of the distribution average” [15].

For the problem in our hands, the observations were the values of the envelope of the seismic trace. We employ the empirical rule in our HEEH technique by assigning a value of one to any envelope value that is greater than three standard deviations from the mean envelope, and a value of zero if it is not. This generates logical windows for the outliers (ones) and zeros. Considering that noise spikes might also generate outliers, we only considered outlier windows which had greater than or equal to four consecutive samples. In general, an outlier is a unique observation that is distinct from other observations in a dataset that contains it. Nevertheless, statistical methods can be used to identify outliers that appear to be rare or unlikely given the available data [16]. The output of an outlier detection command returns a logical array whose elements are those at the locations of the detected outliers and zeroes elsewhere in the corresponding dataset [17], which can be used as the basis for the analysis of the HEEH method.

In an ideal situation, the detected outlier is at the exact location of the first arrival travel time. This is not the case in the Vibroseis record or in data with noise, as some of the noise may exceed three standard deviations from the mean and is presumed to be a signal. Therefore, we see such noise spikes as single events instead of continuous values of ones in the logical array, and we can easily remove them. Unlike noise spikes, the first arrival is not a single event but, rather, a continuous event corresponding to the wavelet. The HEEH retains only windows with ≥ four successive outliers, from which we take the center sample of the first window as the first arrival pick (Figure 3).

## 3. Results

This section discusses the application of the proposed HEEH method to both synthetic and real dataset.

### 3.1. Synthetic Dataset

We used a four-layer acoustic model to calculate the impulse response and generated synthetic data by convolving it with the Klauder wavelet. The four layers had velocities of 800, 2000, 3000, and 4000 m/s, with layer densities of 1648, 2073, 2294, and 2465 kg/m^3^, respectively. The upper three layers had thicknesses of 100, 200, and 300 m, respectively. Direct and head arrival times were calculated using the following Equation (3):(3)$$T_{d}=\;x/v_{1}$$

The following Equation (4) was also used:(4)$$T_{\mathrm{hn}}={\;T}_{0n}+x/v_{n}$$
where x is the offset (spaced at 50 m), v_1_ is the velocity of the first layer, and v_n_ and T_0n_ are the velocity and intercept time of the n^th^ layer for n = 2, 3, and 4, respectively [18]. The Klauder wavelet was generated from the autocorrelation of a linear up-sweep, as in Equation (5), with a minimum frequency of 10 Hz, maximum frequency of 80 Hz, *k* = 7, a sweep length of 10 s, and a sampling interval of 2 milliseconds (ms). The trace was tapered at the edges using a 0.25-s Hanning window before autocorrelation of the following linear up-sweep:(5)$$L\left(t\right)=sin\;\left[2\pi t\left(f{\;}_{min}+kt\right)\right]$$

We then added 10% and 20% Gaussian noise to the clean synthetic dataset to create noisy traces [19]. Figure 4 shows an example trace from the 100 synthetic traces shot record. Figure 5 shows the clean synthetic data and the HEEH picks. Although this figure also indicates the capability of using the HEEH method to pick later arrivals, this subject is beyond the scope of the current research.

We tested the effectiveness of the HEEH method using the absolute error, defined as the absolute difference between the manual pick and the HEEH generated pick. Absolute error tests have been used by many geoscientists (e.g., [20,21]). Figure 6, Figure 7 and Figure 8 show the first arrival picks from the HEEH method versus those calculated from the model for the clean, 10% noise-contaminated, and 20% noise-contaminated synthetic data sets. Despite the increasing amount of noise, the HEEH and calculated picks exhibit an excellent match. Table 1 summarizes the basic statistical parameters of the absolute errors of the synthetic dataset, which show a median absolute error of 0 ms for all datasets, indicating excellent performance. For comparison, Figure 9 shows the absolute errors of all tested synthetic datasets.

### 3.2. Real Dataset

We also tested the proposed HEEH method on Yilmaz shot records 22 and 23, which are Vibroseis records [22]. These shot records contain statics owing to the near-surface complexity. There were 48 traces in each record, with 1650 samples in each trace sampled with a 2 ms sampling rate. Figure 10 shows record 22 before and after picking, using the HEEH method. Figure 11 shows a comparison between the first arrival picks of the HEEH method and those selected manually for this record. Figure 12 shows the absolute errors between the HEEH and the manual first arrival picks of this record. Similarly, Figure 13, Figure 14 and Figure 15 show the results of applying the HEEH method to Yilmaz shot record 23. Because this record has highly noisy traces (numbers 12, 13, and 38), these traces resulted in large absolute error values of up to 0.45 s. Table 2 summarizes the basic statistical parameters of the absolute error for the real dataset. The median absolute error was 10 ms for record 22, indicating a good performance. In comparison, the median absolute error was 18 ms for record 23 because of the presence of a few excessively noisy traces and near-surface complexity.

Furthermore, we calculated the percentage of the number of HEEH picks within 20 ms; the results are shown in Table 3. The chosen threshold value corresponds to accepting picks within half of a 40 ms typical wavelet period ([23,24]). The resulting accuracies for records 22 and 23 were 75% and 67%, respectively, despite a few unpickable traces and the generally low S/N for record 23. In general, the challenging nature of Yilmaz record 23 has been observed in previous studies, which reported considerable inaccuracies without extensive preprocessing. For example, Coppens’ picking method, which is based on energy ratios and their variations, resulted in an accuracy of only 37% within a 20 ms threshold. In comparison, Mousa et al. [23] achieved 41% accuracy after data enhancement using the τ–p transform method.

## 4. Discussion

We proposed and demonstrated the use of the Hilbert transform for first arrival travel time picking of zero-phase seismic data. Earlier attempts that utilized the Hilbert transform in first arrival travel time picking included the MDPE method of Al-Mashhor et al. [25], which was used successfully on minimum-phase data. Another recent paper by Sun et al. [26] applied the empirical formula as a moving window separating the signal into segments, which demonstrated a better denoising effect on non-stationary signals. We tested our method on synthetic and real dataset. We showed that in the case of synthetic data, our method recognizes the first arrival time picking with remarkable accuracy, where the median absolute error from the first arrivals calculated from the depth model was 0 ms. Similarly, the accuracy we observed in the case of two real dataset was 75% and 67%, respectively, within 20 ms from the first arrivals manually picked by an experienced geophysicist.

Although previous methods have demonstrated some robustness, limitations always exist, including error build-up with increasing noise levels, application to a specific source type (minimum phase), optimal window length selection, and the need for human intervention at some stages. Similarly, the performance of the proposed HEEH method was affected by the presence of considerable noise. Despite its limitations for noisy data, the HEEH method is completely automatic and requires no preprocessing or prior parameter testing and/or selection. Furthermore, although the proposed method was tested only on zero-phase data, it is expected to work as well for minimum-phase data by selecting an earlier sample of the first logical window rather than the one in its middle. In addition, we hinted in this paper to the possibility of using the HEEH method for picking later arrivals.

## 5. Conclusions

We introduced a trace-by-trace method for the automatic picking of first arrivals in zero-phase seismic data, based on trace envelope calculation, outlier detection, and first arrival selection. These simple yet effective steps produce good results quickly and with high accuracy, even with low S/N data. The validity of our method was tested on both synthetic and real seismic datasets. Tests on clean and noisy zero-phase synthetic datasets with multiple first arrivals (i.e., direct and three head waves) showed an excellent accuracy of over 99% between the picked and calculated first arrivals. Furthermore, the HEEH method was tested on Yilmaz shot records 22 and 23, which had a Vibroseis source. The proposed method was able to pick first arrivals with a median absolute error of only 10 ms (i.e., five samples) on record 22. In comparison, the method’s testing on record 23 resulted in a median absolute error of 18 ms (nine samples) owing to the presence of excessive noise, particularly on a few scattered traces.

An advantage of the proposed HEEH method is that it is completely automatic, requiring no user intervention. In addition, it can be easily implemented because its main steps are available in commonly used seismic and signal processing software packages. Furthermore, it does not require any preparation of the data. On the other hand, similar to most first arrival picking techniques, the performance of the HEEH method deteriorates with the decreasing S/N ratio, as seen from tests on the low S/N ratio Yilmaz shot records 22 and 23.

## Figures and Tables

**Figure 1 sensors-22-07580-f001:**
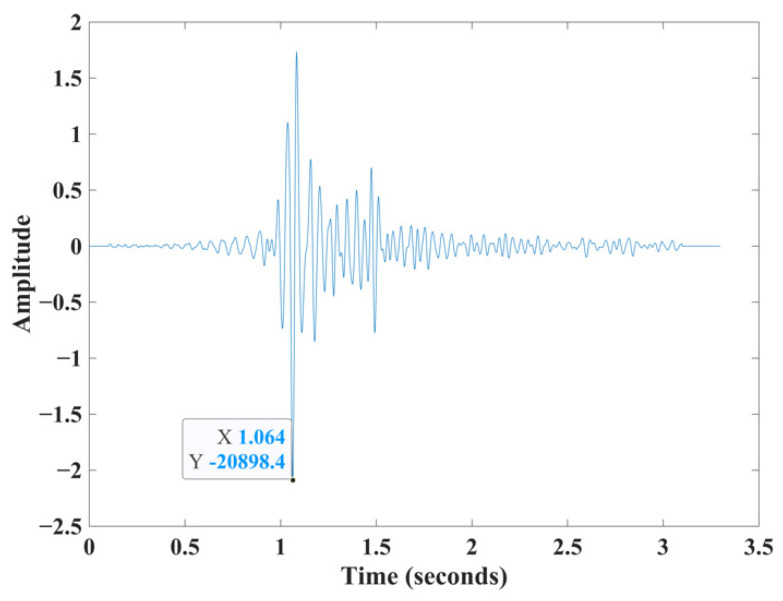
The first arrival travel time manually picked (at time 1.064 s) on a real Vibroseis trace to illustrate the HEEH method.

**Figure 2 sensors-22-07580-f002:**
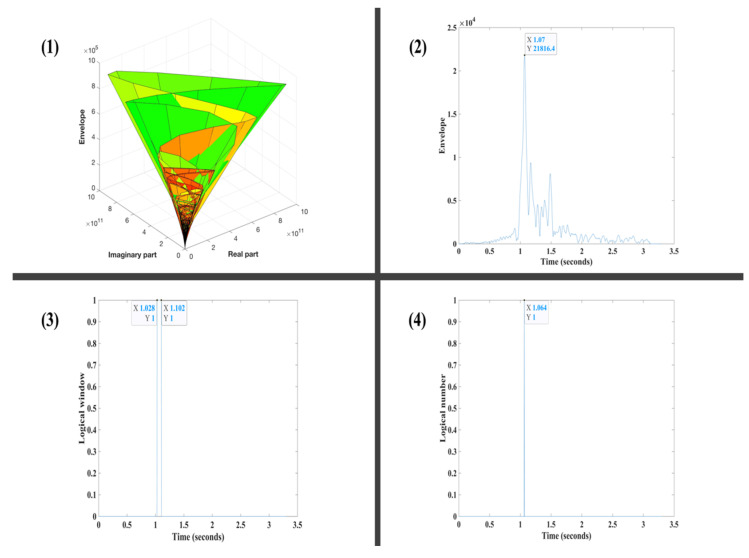
Steps 1–4 of the HEEH method applied to a real Vibroseis trace: (**1**) The Hilbert transform of the trace in Figure 1 results in a complex seismic trace. (**2**) The envelope step amplifies the onset of the first arrival travel time location with a noticeable jump relative to preceding samples. The location of the maximum envelope value is indicated (the maximum value in the envelope step above is not always the first break, as shown in Figure 1). (**3**) Outlier detection step uses the empirical rule to provide logical windows of ones and zeros. The start and end points of the resulting outlier logical window are indicated. (**4**) The half-window step selects the midpoint of the first outlier window in step 3 as the first arrival time. Compare this pick with the manually picked arrival in Figure 1.

**Figure 3 sensors-22-07580-f003:**
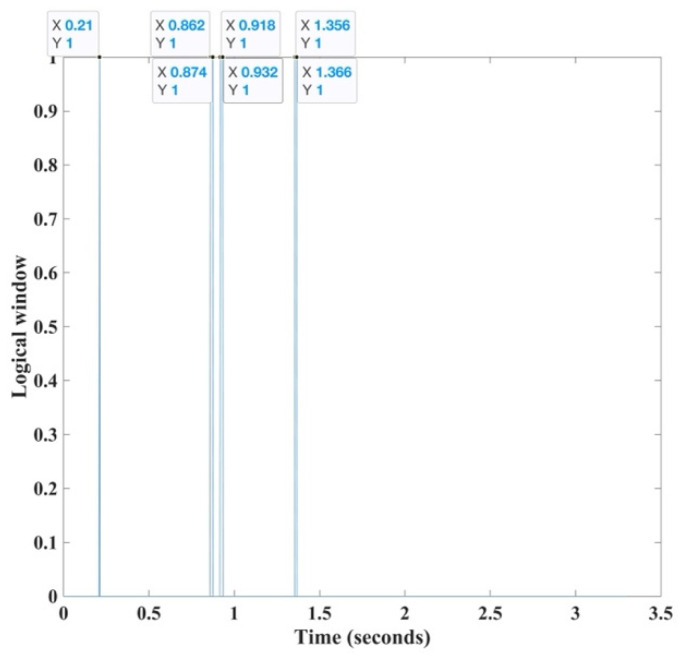
The outlier detection window (applied to trace 45 of Yilmaz shot record 23) is a logical array of ones and zeros. The first outlier (at time 0.21 s) is noise as it is a window of one sample only (thus ignored). The second window (at time 0.862–0.874 s) is an array of ones and it is our first arrival window. The third (at time 0.918–0.932 s) and fourth (at time 1.356–1.366 s) windows nearly match the size of the first arrival window but come later and, thus, are defined as later arrivals.

**Figure 4 sensors-22-07580-f004:**
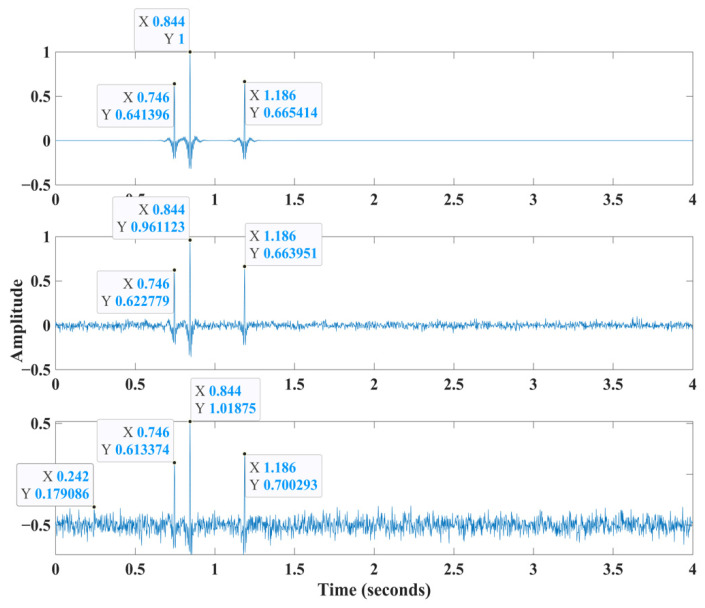
Clean synthetic data example trace number 20 (**top**) with the location of the first and later arrivals indicated. After adding 10% noise to the same synthetic trace (**center**). After adding 20% noise to the same synthetic trace (**bottom**).

**Figure 5 sensors-22-07580-f005:**
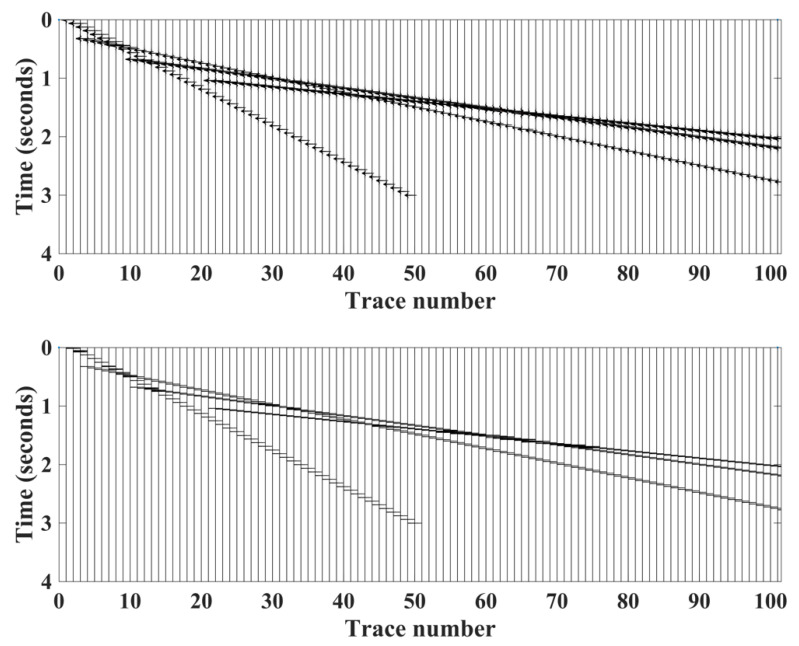
Clean synthetic data (**top**) with the locations of the first and later arrival picks generated using the HEEH method (**bottom**). The direct arrival is indicated by the linear event that has an intercept of zero, while the other linear events indicate head waves from various layer interfaces.

**Figure 6 sensors-22-07580-f006:**
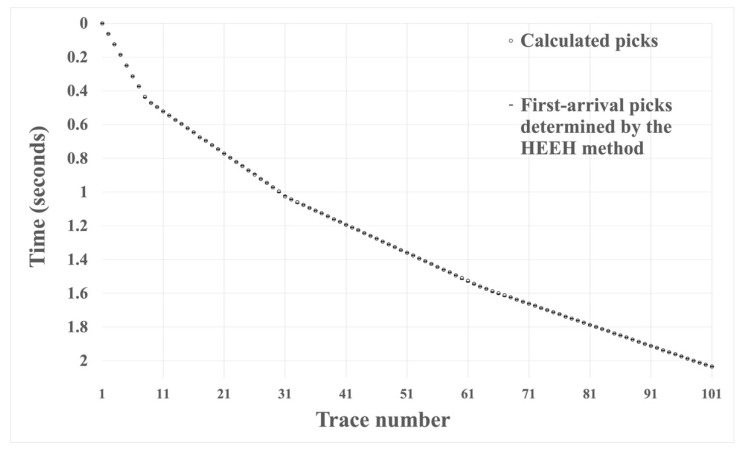
First arrival picks determined by the HEEH method and calculated from the model on the clean synthetic data.

**Figure 7 sensors-22-07580-f007:**
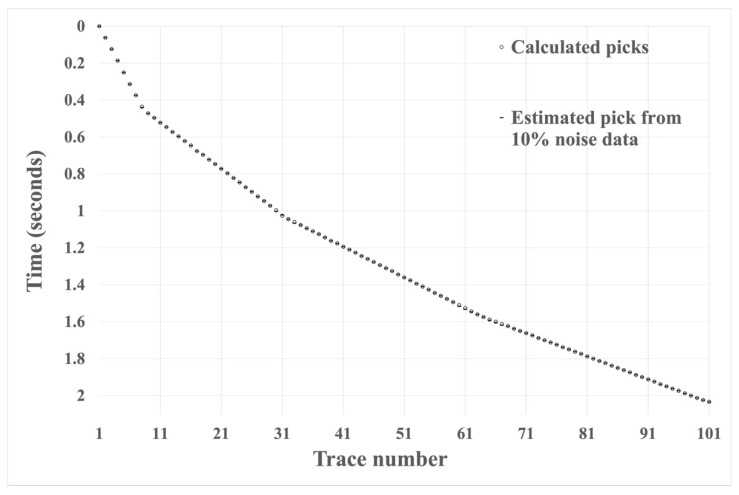
First arrival picks determined by the HEEH method and calculated from the model on synthetic data with 10% noise.

**Figure 8 sensors-22-07580-f008:**
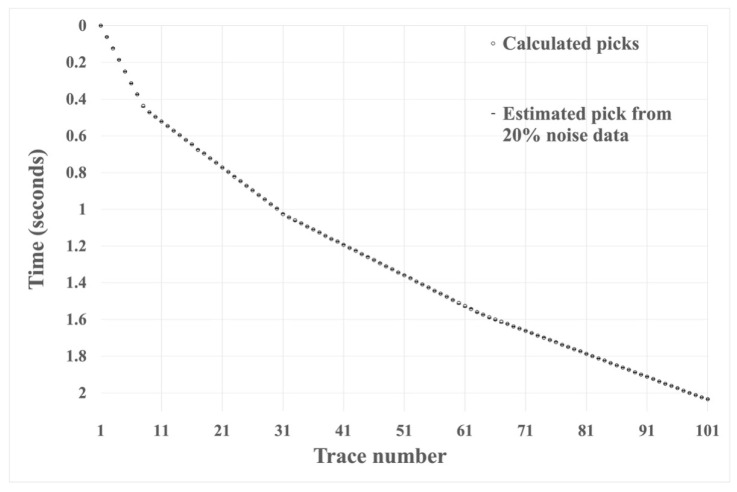
First arrival picks determined by the HEEH method and calculated from the model on synthetic data with 20% noise.

**Figure 9 sensors-22-07580-f009:**
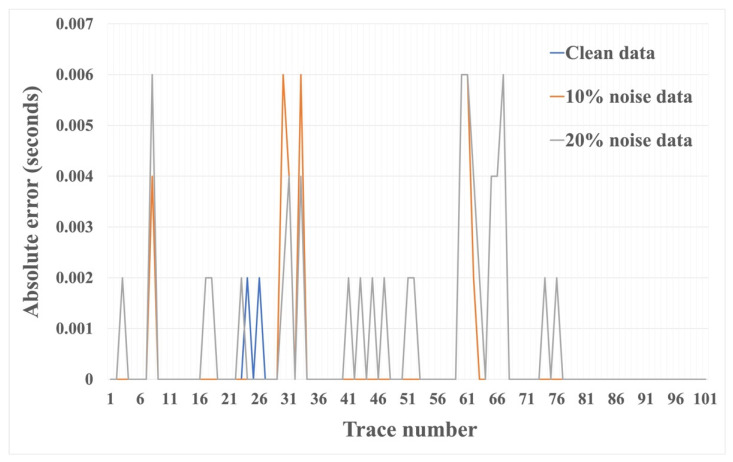
Absolute error between first arrival picks determined by the HEEH method and calculated from the model on the synthetic dataset.

**Figure 10 sensors-22-07580-f010:**
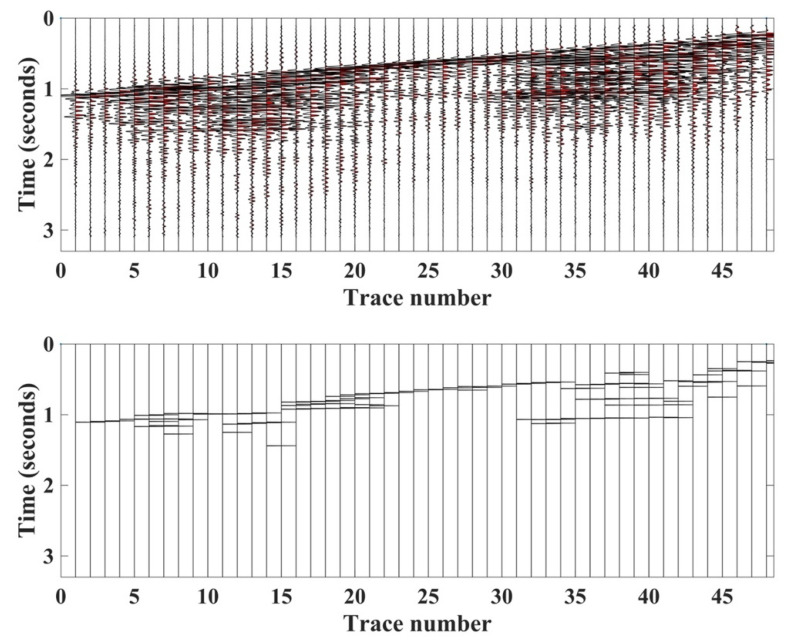
Yilmaz record number 22 (**top**) and picks determined by the HEEH method (**bottom**).

**Figure 11 sensors-22-07580-f011:**
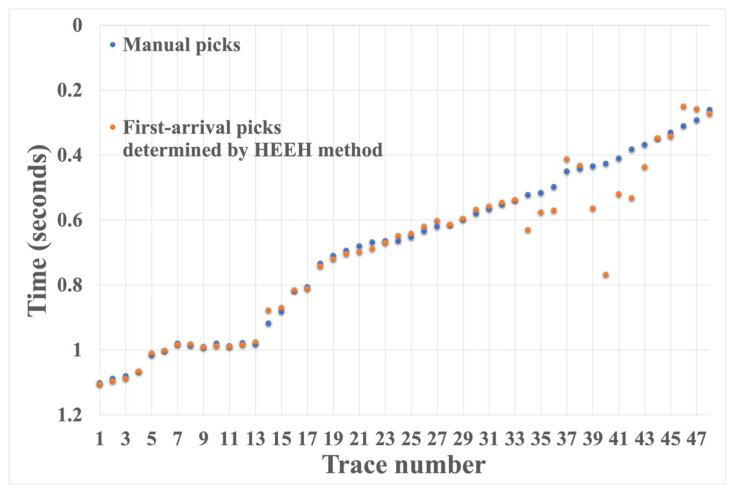
Yilmaz record number 22 first arrival picks determined by the HEEH method and manually.

**Figure 12 sensors-22-07580-f012:**
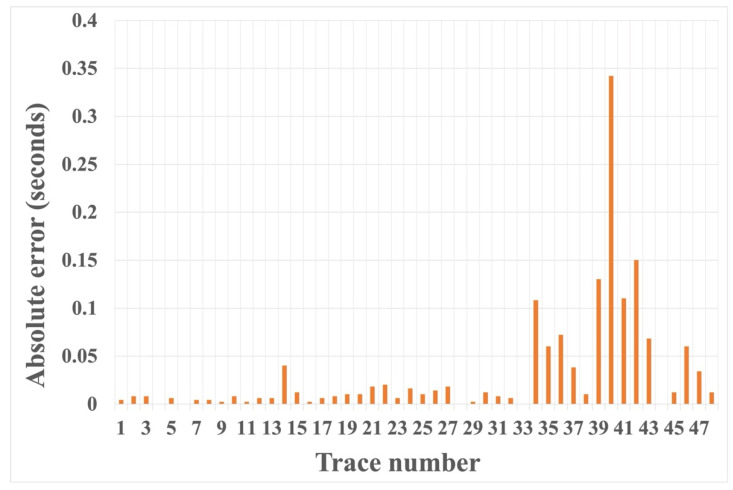
Absolute error between first arrival picks determined by the HEEH method and manually for Yilmaz record number 22.

**Figure 13 sensors-22-07580-f013:**
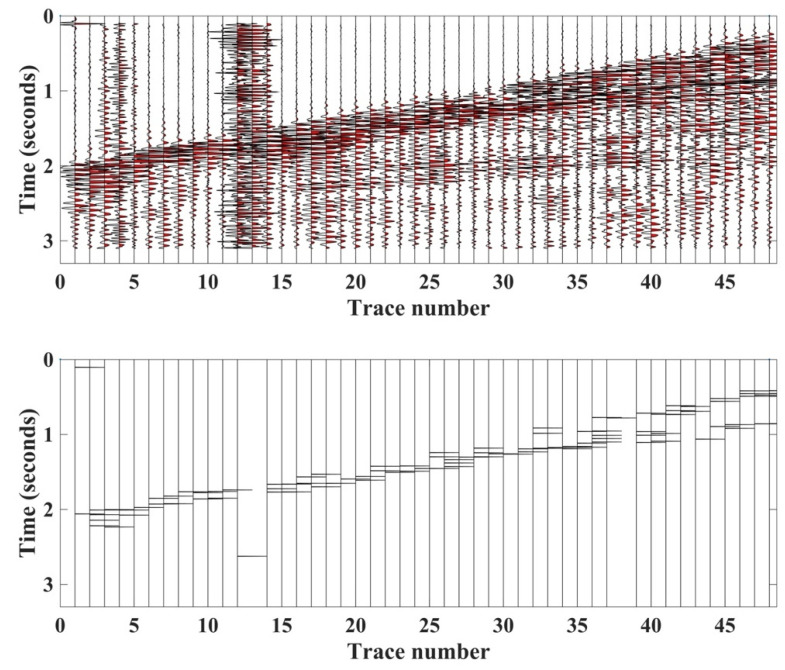
Yilmaz record number 23 (**top**) and the estimated picks using the HEEH method (**bottom**).

**Figure 14 sensors-22-07580-f014:**
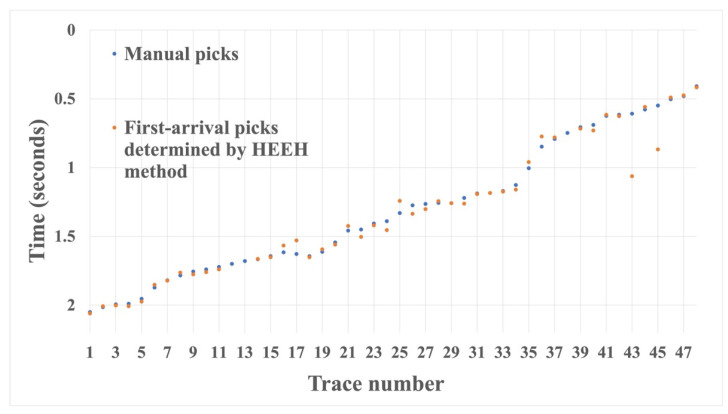
Yilmaz record number 23 first arrival picks determined by the HEEH method and manually.

**Figure 15 sensors-22-07580-f015:**
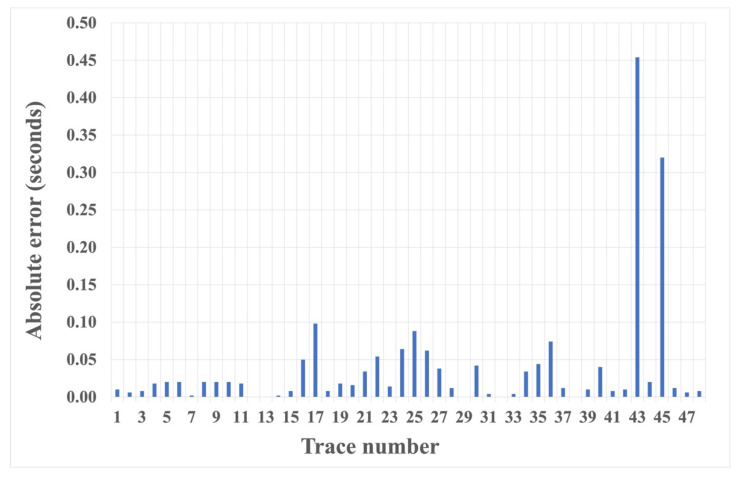
Absolute error between first arrival picks determined by the HEEH method and manually for Yilmaz record number 23.

**Table 1 sensors-22-07580-t001:** Basic statistical parameters of the absolute error values (in seconds) for the synthetic dataset.

Data Type	Minimum Error	Maximum Error	Median	Average	Standard Deviation
Clean synthetic data	0	0.06	0	0.0005	0.00151
Synthetic data with added 10% noise	0	0.06	0	0.0004	0.00150
Synthetic data with added 20% noise	0	0.06	0	0.0007	0.00151

**Table 2 sensors-22-07580-t002:** Basic statistical parameters of the absolute error values (in seconds) for the real dataset.

Data Type	Minimum Error	Maximum Error	Median	Average	Standard Deviation
Shot record 22	0	0.342	0.010	0.030	0.058
Shot record 23	0	0.454	0.018	0.040	0.080

**Table 3 sensors-22-07580-t003:** The percentage of the number of HEEH picks within 20 ms for the real dataset.

Data Type	|Manual-HEEH| ≤ 20 ms	Non-Picked Traces
Shot record 22	75%	0
Shot record 23	67%	3

## Data Availability

Not applicable.

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
