# Peer review of "First Arrival Picking of Zero-Phase Seismic Data by Hilbert Envelope Empirical Half Window (HEEH) Method"

_sensors, 2022, doi:10.3390/s22197580_

Round 1

Reviewer 1 Report

Dear colleagues, very good work very interesting.

Unfortunately I have to ask you to make the latest changes. The Material and Methods paragraph is poorly organized, there are contents but text and figures do not speak to each other. A person out of the business such as a geophysical trained seismologist would have a hard time understanding you. The contents and the figures are there, I think it is better to simply add an introduction figure in which all the passages are shown and then start with the details trying to bring the figures closer to the part of the text concerned. Above all, I do not recommend subdivision into subparagraphs. It gives the feeling that the topics are not linked to each other, confusing a possible colleague who does not have a more unbalanced training on other fields.

The rest of the manuscript is perfect, even if the conclusions are very short, yet the discussion is handled well.

I suggest including a short remake of the discussions in the conclusion.

Regards

Reviewer

Author Response

Response to reviewers’ comments of manuscript number

sensors-1906894

Response to reviewer 1:

Dear colleagues, very good work very interesting.

  1. Unfortunately I have to ask you to make the latest changes. The Material and Methods paragraph is poorly organized, there are contents but text and figures do not speak to each other. A person out of the business such as a geophysical trained seismologist would have a hard time understanding you. The contents and the figures are there, I think it is better to simply add an introduction figure in which all the passages are shown and then start with the details trying to bring the figures closer to the part of the text concerned. Above all, I do not recommend subdivision into subparagraphs. It gives the feeling that the topics are not linked to each other, confusing a possible colleague who does not have a more unbalanced training on other fields.
  2. The rest of the manuscript is perfect, even if the conclusions are very short, yet the discussion is handled well.
  3. I suggest including a short remake of the discussions in the conclusion.

 Regards

Reviewer

Thanks to the reviewer for his encouraging and constructive comments. Our responses are outlined below:

  1. In response to this comment, Figures 2-5 of the introduction have been combined into one figure.
  2. We expanded the conclusion section to include a new paragraph on the main advantages and limitations of the proposed method.
  3. We highlight some parts of the discussion section in the conclusion section in the revised manuscript.

Reviewer 2 Report

The methodology proposed is certainly very interesting and looks promising, however much more testing on real cases is needed to prove its validity.

Author Response

Response to reviewers’ comments of manuscript number

sensors-1906894

Response to reviewer 2:

The methodology proposed is certainly very interesting and looks promising, however much more testing on real cases is needed to prove its validity.

We agree with the reviewer. However, for the sake of brevity, we conducted tests of the HEEH method on only samples of synthetic and real datasets. For synthetic data (even with a high noise level of 20%), the HEEH method showed a remarkable accuracy of 99%. However, tests on real data with relatively low S/N ratio resulted in a lower, yet acceptable, accuracy.

Reviewer 3 Report

The manuscript is interesting but in the current form it is not possible to accept.

1. English is to be improved.

2. The introduction is to be rewritten with some more recent references.

3. Figures are of low resolution.

4. Discussion is not sufficient. Still to be improved. 

1. What is the role of thickness and density in 4-layer acoustic model? 2. How is the noise % connected to amplitude? 3. Explain the First arrival picks determined by the HEEH method.

Author Response

Response to reviewers’ comments of manuscript number

sensors-1906894

Response to reviewer 3:

  1. What is the role of thickness and density in 4-layer acoustic model?
  2. How is the noise % connected to amplitude?
  3. Explain the First arrival picks determined by the HEEH method.

  1. The thickness affects the first arrival time through the intercept time (T0n) in eq. (4). For example, the intercept time for the first head wave (from the interface between layers 1 and 2) is given by the following relation (Sheriff and Geldart, 1995):

T01=(2H1/V1) [Cos(qc1)],

where H1 is the thickness of layer 1 and qc1 is the critical angle at the interface between layers 1 and 2. The densities do not affect the first arrival time and we included them only for the sake of model completion.

  1. The noise % affects the amplitude as follows:

An(i)=A(i)(1+ni),

where A(i) and An(i) are the amplitudes at the i-th sample before and after adding noise and ni is the amount of noise added at the i-th sample.

  1. As explained in Figure 2 of the revised manuscript, the first arrival pick is determined by the HEEH method through:
    1. calculating the trace envelope
    2. searching for the first window of outliers in the trace envelope
    3. assigning the first arrival pick of this trace at the middle of this window.

Round 2

Reviewer 2 Report

 thank you for adding more testing